# Thermal Degradation Kinetics of ZnO/polyester Nanocomposites

**DOI:** 10.3390/polym12081753

**Published:** 2020-08-05

**Authors:** E. A. Franco-Urquiza, J. F. May-Crespo, C. A. Escalante Velázquez, R. Pérez Mora, P. González García

**Affiliations:** 1CONACYT—CIDESI, Centro Nacional de Tecnologías Aeronáuticas (CENTA). Carretera Estatal 200 Querétaro-Tequisquiapan km 23 No. 22547, Colón 76270, Querétaro, Mexico; edgar.franco@cidesi.edu.mx (E.A.F.-U.); carlos.escalante@cidesi.edu.mx (C.A.E.V.); ruben.perez@cidesi.edu.mx (R.P.M.); 2CONACYT—El Colegio de Michoacán, Cerro de Nahuatzen 85, Fracc. Jardines del Cerro Grande, La Piedad 59370, Michoacán, Mexico; jmayc1979@gmail.com

**Keywords:** unsaturated polyester resin, nanocomposites, thermal degradation kinetics, ZnO

## Abstract

ZnO particles were synthetized by the sol–gel method and subsequent heat treatment of 400, 500 and 600 °C was applied. The nano ZnO particles were incorporated to the unsaturated polyester resin by solution blending at 0.05 wt % concentration. X-ray diffraction detected the formation of a wurtzite-like structure. Viscoelastic behavior of neat polyester and nanocomposites revealed the nano ZnO particles does not promote better mechanical properties because of a weak interaction and the glass transition temperature of the polyester was favored by the presence of a higher quantity of nano-size ZnO particles. Thermogravimetric analysis at 5, 10 and 20 °C/min allowed determining the degradation kinetic parameters based on the Friedman and Kissinger models for neat polyester and nanocomposites. Heating rates promoted an increase in the temperature degradation and the addition of ZnO particles promoted a catalyst effect that reduce the amount of thermal energy needed to start the thermal degradation.

## 1. Introduction

Polymer nanocomposites (PNC) are attractive to research and industrial interests due to their potential for versatile applications. They consist of a polymer matrix that contains nanofillers or nanostructured particles uniformly dispersed. The transition from the micro to nanoscale depends on the characteristics of the fillers such as the crystal structure, concentration, dispersion, size and geometry (platelet, sheet, tape, flake, bar, tube or fiber), which leads to a change in mechanical and thermal properties of the neat polymers [1,2].

Metal oxides (RuO_2_, ReO_3_, TiO_2_, BaTiO_3_ and ZnO) represent a growing asset in many industries, especially because of their chemical, physical and electronic properties. They are considered as heterogeneous catalysts and applied for acid–base and redox reactions. Furthermore, metal oxide nanoparticles are widely used in various reactions, which include oxidation, dehydration, dehydrogenation and isomerization, reasons they are versatile materials that can be used in applications such as medical technology, energy, water treatment and personal care products among others [1,3,4].

Particularly, zinc oxide (ZnO) is a promising candidate for different applications because of its intrinsic properties depending synthesis conditions [5]. ZnO crystallizes in the wurtzite (hexagonal), zinc blende (cubic) and rock salt (cubic) phases, being the wurtzite phase the most stable structure at room temperature [6]. 

The ZnO wide band gap and its high thermal stability lead to a wide range of properties reported in terms of UV resistance [6], antimicrobial evaluation [7], hydrophobicity efficacy [8] and enhanced mechanical properties [1,9].

ZnO nanostructures can be synthesized by different processes such as chemical vapor deposition (CVD) [10], supercritical fluid deposition (SCFD) or in situ method. However, the first two processes deal with surface treatment, where bulk properties like mechanical or thermal cannot be achieved [11]. Further, nanoparticles produce aggregation because of lacks high yield and control on the morphology of nanoparticles [12].

ZnO nanoparticles have been extensively used in the development of the PNCs in both the thermoset and thermoplastic polymer matrix, and their applications in the field of packaging, medicine and textile have been reported [13].

Unsaturated polyester resin is a thermoset polymer widely used to impregnate glass fibers for the transport sector (automotive, railway and marine) because of its low cost, density, good corrosion resistance and high strength-to-weigh ratios [14]. Nonetheless, thermal degradation and mechanical performance of polyester resins limit their application to other relevant industrial sectors like aerospace.

Kinetic studies are commonly used to optimize the curing process of composites. The kinetics of the thermal behavior of unsaturated polyester can be determined by the use of kinetic model to the rates of mass degradation. Thermogravimetry analysis is useful in order to explain kinetics of thermal decomposition as qualitative information.

There are several methods for evaluating kinetic parameters through non isothermal thermogravimetric analysis, being the most common the model-fitting and the model-free [15,16,17,18]. The model-fitting approach employs different models to fit the experimental data. The best statistical fit is selected as the model from which the activation energy and frequency factor are evaluated. The model-free approach is known as isoconversional method due to its ability in determining the activation energy without considering any particular form of the reaction model, and it is able to calculate the activation energy at different heating rates on the same value of conversion.

It is relevant to highlight the terms “model-free” and “isoconversional” are regular used interchangeably. However, not all model-free are isoconversional analysis. The typical case is the Kissinger model, which is a model-free but is not the isoconversional approach because it does not calculate activation energy at different constant extents of conversion but instead assumes constant activation energy [15,18,19]. The Friedman model is another isoconversional approach, just this model is a derivative method (Kissinger is integral method) and perhaps the most general of the derivative techniques [18,20].

The aim of this work is to present the effect of the heat treatment applied to ZnO particles on the mechanical and thermal properties of unsaturated polyester resin (UPR). The thermal degradation kinetics based on the Friedman and Kissinger models for neat UPR and UPR/ZnO nanocomposites is also evaluated.

## 2. Materials and Methods

### 2.1. Materials

The UPR PP-250 provided by Poliformas Plasticas Mexico was used as a matrix. This liquid resin is composed of a copolymer of maleic acid (30% mol) and isophthalic acid (30% mol) with propanediol (40% mol), and styrene (20 wt % styrene) with a small fraction of polymerization inhibitor. Since the PP-250 resin is hardened via a radical process, cobalt octoate used as an accelerator and the methylethylketone peroxide initiator solution were added to the mixture.

Analytical grade zinc acetate (Zn(CH_3_COO)_2_·2H_2_O 99.8%) and oxalic acid (C_2_H_2_O_4_·2H_2_O, 99.8%) from Sigma-Aldrich were used without further purification.

### 2.2. Synthesis of ZnO Nanoparticles

The sol–gel method was used for the synthesis of ZnO nanoparticles. Zn(CH_3_COO)_2_·2H_2_O and C_2_H_2_O_4_·2H_2_O mixtures were prepared in ethanol with a molar ratios of 0.1:0.1. Oxalic acid and ethanol were mixed at 50 °C for 30 min in a hot plate magnetic stirrer JoanLab SH-4. The zinc acetate was heated at 60 °C for 30 min in a circulating water bath Thermo Scientific™ Precision™ TSCIR19. The oxalic acid solution was subsequently incorporated dropwise into the zinc acetate solution under vigorous stirring until a viscous gel was formed, being stirred continuously for additional 90 min. The operating pH of 2.0 was measured with a pH meter and obtained by initial addition of oxalic acid to the mixture.

Afterwards, heat treatment on the ZnO particles at isothermal conditions of 400, 500 and 600 °C under air atmosphere for 3 h was performed.

### 2.3. UPR/ZnO Nanocomposites 

ZnO nanoparticles were incorporated by solution blending into the polyester matrix at a concentration of 0.05 wt % in order to obtain UPR/ZnO400, UPR/ZnO500 and UPR/ZnO600 nanocomposites.

ZnO nanoparticles were suspended in ethanol and ultrasonicated for 5 min before the solvent was partially evaporated by heating the dispersion at 50 °C. Then, UPR was added to the dispersion, being mixed and mechanically stirred by hand until ethanol was completely evaporated and an apparent uniform distribution of the ZnO in the resin was achieved.

In addition, the mixture was placed in silicone molds and cured in an oven at 60 °C for 12 h in order to obtain rectangular specimens of 80 mm × 13 mm × 3 mm.

### 2.4. Characterization Techniques 

X-ray diffraction (XRD) patterns of ZnO powders heat treated at 400 and 600 °C were recorded using a Rigaku D/max-2100 diffractometer (Cu kα radiation, 1.5406 Å) in the range of 20–75° for an incidence angle of 0.5°. 

The structural analysis of the polyester resin and its nanocomposites were carried out using a Fourier-transform infrared spectroscopy (FTI-IR) Perkin Elmer Frontier. Rectangular molded specimens were placed in the MIR-ATR optics assembly of the instrument. FTIR spectra were recorded by scanning the samples in the range of frequency 4000–500 cm^−1^. The specimens were scanned at 32 scan times in the transmittance mode at a resolution of 4 cm^−1^.

The influence of the ZnO heat treated nanoparticles on the viscoelastic properties of UPR was evaluated in a dynamic mechanical analyzer, Perkin Elmer DMA 800. The DMA parameters such as storage modulus (E′) and loss tangent (Tan δ) were determined using the 3 point bending configuration at a frequency of 1 Hz. The specimens with dimensions of 80 mm × 12.5 mm × 2.7 mm were tested at a heating rate of 10 °C/min in the range of 40–160 °C. The support span was fixed at 43 mm with a loading point at the middle length of the specimens.

The thermal stability and pyrolysis were characterized in a thermogravimetric analyzer Perkin Elmer TGA 4000 (Beaconsfield, UK). TGA dynamic experiments were performed at three different heating rates of 5, 10 and 20 °C/min from 40 to 600 °C. The furnace was continuously purged with inert nitrogen gas atmosphere at flow rate of 20 mL/min to displace air from the pyrolysis zone by avoiding unwanted oxidation of the sample.

### 2.5. Kinetic Models

The degradation behavior of polymers can be studied in terms of the partial mass loss, which is defined by:(1)$$\alpha =\frac{W_{0}-W_{t}}{W_{0}-W_{f}}$$
where $W_{0}$, $W_{t}$ and $W_{f}$ represent the initial, actual and final mass of the material, respectively. 

Thermal degradation rate of a polymer [19] can be described by: (2)$$\frac{d\alpha }{\mathrm{dt}}=k\left(T\right)f\left(\alpha \right)$$
where α corresponds to the fraction of solid degraded at time t, f(α) is the function that depends on the reaction mechanism and k is the rate constant given by the Arrhenius equation [20,21]: (3)k(T)=A exp(−EaRT)
where A is the frequency factor (1/min), E_a_ is the activation energy (J/mol), R is the universal constant of gases (8.314 J/mol K) and T is the reaction temperature (K). Substituting the Equation (2) into the Equation (1), and introducing the constant of heating rate β (β = dT/dt, °C/min), the decomposition degree can be written as:(4)dαdT=Aβ exp(−EaRT)f(α)

Other models have been proposed in order to describe the reaction mechanism, one of them is the Friedman analysis [20], which is based on the assumption that the rate of conversion is proportional to n-th order of the material concentration, as follows: (5)$$f\left(\alpha \right)={\left(1-\alpha \right)}^{n}$$

Substituting Equation (4) into Equation (3) and calculating the natural logarithm, the following equation is obtained: (6)$$ln\left(\;\frac{d\alpha }{dT}\;\right)=lnA+nln\left(1-\alpha \right)-\frac{E_{a}}{RT}$$

The degradation kinetic parameters E_a_ and n can be calculated from the Friedman equation as follows: the plot of ln(dα/dt) versus 1/T for each α value is obtained directly from thermograms at different heat rates. According to the Equation (6) this curve should be a straight line whose slope allows one to calculate the corresponding E_a_ value. In a similar manner, the corresponding value of order of the reaction n, can be obtained from the slope of the plot ln(1 − α) versus 1/T. In order to facilitate the calculations, the values of α = 0.2, 0.4, 0.6 and 0.8 and their corresponding values of ln(dα/dt), ln (1 − α) and 1/T were considered for the fitting analysis that allow one to calculate the corresponding E_a_ and n constants. Friedman method is frequently used to evaluate the thermal degradation kinetics because of its simplicity and it does not involve additional approximations, but it requires numerical differentiation (see Equation (6)), which introduces some inaccuracy in the results of the kinetic parameters. That is the main reason why the integral isoconversional methods, such as the Kissinger method is regarded a good alternative to determine the degradation kinetic parameters. Kissinger rate equation is expressed as follows [20,21]:(7)$$ln\left(\;\frac{\beta }{T_{max}^{2}}\;\right)=ln\frac{AR}{E_{a}}+ln\left(n{\left(1-\alpha_{max}\right)}^{n-1}\right)-\frac{E_{a}}{RT_{max}}$$
where T_max_ and α_max_ are the temperature and conversion at the maximum conversion rate (dα/dt)_max_ for different heat rates β. In this case the activation energy was obtained from the slope of the linear fit of the plot $ln\left(\frac{\beta }{T_{max}^{2}}\right)$ versus $\frac{1}{T_{max}}$. The interception $I=ln\frac{AR}{E_{a}}+ln\left(n{\left(1-\alpha_{max}\right)}^{n-1}\right)$ of this linear plot allows to compute the order of reaction n, as follows [20]:(8)n=(1−αmax)[Exp(−EaRTmax)]R(dαdt)Ea(Exp(I))

The Coats–Redfern method is frequently used to analyze the solid-state mechanism of thermal degradation, this method is based on an approximation of the integral form of the Equation (3) according to:(9)$$\ln\left(\frac{g\left(\alpha \right)}{T^{2}}\right)=ln\left(\frac{AR}{\beta E}\right)-\frac{E_{a}}{RT}$$

The function g(α) depends on the degradation mechanism and several theoretical functions are proposed, a compilation of algebraic functions are presented for other authors [21,22]. In this work, the activation energy for decomposition was calculated from the slope of the plot of ln(g(α)/T^2^) vs. 1/T.

## 3. Results and Discussion

### 3.1. ZnO Microstructure

X-ray diffraction patterns of ZnO particles heat treated at 400 and 600 °C are presented in Figure 1. The presence of ten reflections in 2θ from 20 to 75°, pointed out to the formation of a crystalline material; their position (according to the card JCPS #36-1451 used as a reference) indicated that the formed material actually was a hexagonal wurtzite phase of ZnO. The sharp reflections indicate the good crystallinity of the prepared particles, and the peak broadening the presence of small nanocrystals. It is easy to note the reflections displayed for the ZnO particles treated at 600 °C are sharper and more defined than the particles treated at 400 °C, which indicates an increase in the crystallinity by applying the thermal process. A similar result was analyzed for other authors [22]. Bindu and Thomas [23] remarked that as ZnO crystallizes in the wurtzite structure, in which the oxygen atoms are arranged in a hexagonal close packed type with zinc atoms occupying half the tetrahedral sites, the Zn and O atoms are tetrahedrally coordinated to each other, hence they have equivalent positions.

Furthermore, both XRD of the heat treated ZnO nanoparticles present broad reflections located at 2θ ≈ 31.8, 34.4, 36.3, 47.5, 56.6 and 62.8° that were assigned to the (100), (002), (101), (102), (110) and (103) crystalline planes of the ZnO in the wurtzite-type structure [24,25].

The particle size (D) can be estimated from the full width at half maximum (FWHM) of the peaks in the XRD patterns using the Debye–Scherrer equation, as expressed in the following equation:(10)$$D=\frac{K\lambda }{\beta cos\theta }=\frac{0.9\lambda }{Bcos\theta }$$
where β is the full width at half maximum of the experimental diffraction signals expressed in radians, θ is the half diffraction angle of the peak—in the case of graphene or layered silicates this corresponds to interlayer spacing—λ is the wavelength (1.5418 Å), and K is a constant related to crystallite shape [26]. The average particle size calculated in this work was of 38 and 19 nm for the ZnO nanoparticles heat treated at 400 and 600 °C respectively.

### 3.2. FT-IR Spectroscopy

FTIR spectra of the UPR and UPR/ZnO nanocomposites are shown in Figure 2. The FTIR spectra were very similar between UPR and nanocomposites as observed in Figure 2a.

The UPR spectrum was used as a reference to confirm the presence of characteristic bands compositions. Since unsaturated polyester may contain two or more different sources of aromatic residues, being the most commonly styrene, the most characteristics spectra should be the strong carbonyl stretching band [27].

The characteristic peaks of UPR located in the wavenumber range of 3020–2853 cm^−1^ are attributed to saturated aliphatic (alkane/alkyl) group frequencies [4].

The strong band at 2919 cm^−1^ was prominent and sharper, however its intensity was reduced by the presence of ZnO nanoparticles. The higher heat treatment on ZnO the lower intensity of signals. It seems the heat treatment on ZnO fillers was promoting the presence of –CH=CH– groups, which are responsible of the bands that almost disappeared in the IR spectrum of nanocomposites. According to Bharat Dholakiya, the previous could be related to the presence of the alkane group that contributes with the –CH=CH group through the curing process and conversion of this group to alkane during the cross linking process. Furthermore, other authors [26,28] explained the vibrational bands lead to disappearance due to the presence of absorbed groups on the surface of ZnO nanocrystals.

The bands at 1721 cm^−1^ and 729 cm^−1^ fit well to the functional groups with characteristic bands compositions such as anhydride and aromatic compounds, respectively [24]. The former confirms the presence of –C=O stretching in the ester group, meanwhile the latter is responsible of the C-H bending arising from the first and third position in the benzene ring.

The strong band identified at 1721 cm^−1^ should also confirm the formation of the polyester resin. Alkane –CH2– was confirmed by the presence of broad-spectrum band at 1453 cm^−1^ and it was originated from the C–H bending vibrations (deformation) in CH_3_ and CH_2_. Furthermore, the medium band located at 1229 cm^−1^ emerged from the C–O bond stretching.

Two asymmetric bands at 1149 and 1074 cm^−1^ are characteristics for oxygen containing groups –C–O–C– of ester linkages [29].

Close observations in Figure 2b allowed us to appreciate relevant bands present in nanocomposites at 668 cm^−1^. The signals are assigned to the stretching vibrations of Zn-O bonds in octahedral arrangements and further confirms the wurtzite structure [30,31].

### 3.3. Dynamic Mechanical Analysis

The dynamic mechanical results expressed as storage modulus (E’) and loss tangent (tan δ) as a function of temperature for the UPR and UPR/ZnO nanocomposites are shown in Figure 3. The storage modulus, commonly associated with the Young’s modulus, was measured from the end of glassy region up to the rubbery plateau (Figure 3a).

It is possible to appreciate that the presence of ZnO nanoparticles reduced the storage modulus of the polyester resin. Moreover, neat polyester had a stiffness of 4935 MPa, while the storage modulus values decreased to 3860, 3190 and 2700 MPa for nanocomposites containing ZnO particles heat treated at 400, 500 and 600 °C, respectively.

The drop in storage modulus could be related to the weak interaction between the polymer chain structure and filler. Furthermore, the heat treatment produced an effective size reduction of the ZnO particles promoting an increase of the surface area. Hence, as the surface area/volume ratio increased with the decrease in the size of the nanoparticle, this might lead to the low stiffening effect [3,32,33].

Note the curve corresponding to the neat UPR shows two distinct behaviors with respect to the nanocomposites (Figure 3a). The first one is observed in the transition region, where the polyester curve drops abruptly with a pronounced slope, which differ to the nanocomposites where the transition zone is less pronounced. The second one is appreciated in the rubbery modulus region, where the polyester develops a notorious shoulder from 90 to 160 °C, afterwards the curve tends to zero. Since this signal is related to the crosslinking density, which is activated by effect of temperature, this kind of shoulder should be related to the partial curing process [3,34,35]. The behavior developed by the UPR allows one to consider that additional curing of the unsaturated polyester is occurring.

After the modulus inflection from 92 °C, the thermal energy provided sufficient molecular mobility to restart the curing process of the polyester resin, causing an increase in the storage modulus with a maximum peak located at 116 °C. As the temperature further increased the curing reaction decreased and the system approached the full cure, when observing the storage modulus it decreased again and tended to reach zero within the rubbery region [36].

In the case of the nanocomposites, the rubbery plateau tended to reach zero after the transition region, which should imply that the ZnO nanoparticles favor the curing process of the polyester resin [3,34,35], which is in agreement to the FTIR results previously shown.

The mechanical loss factor as a function of temperature is presented in Figure 3b.

It is possible to appreciate that the neat UPR revealed two transition peaks denoted as α and α’, located at 89 and 144 °C respectively.

The α relaxation reflects the glass transition temperature of the crosslinking structure, hence this signal is greatly dependent on the degree of crosslinking of the UPR, which is governed by the styrene content. The following α’ signal is assigned as a secondary relaxation associated to the segmental motions of long chains of the polyester backbone and promoted by the uncured chains. In other words, the α signal corresponds to the fully cured material and the crosslinking density, meanwhile α’ is related to the uncured system [37]. UPR nanocomposites revealed just one tang δ signal, which represents that the DMA tests were run in complete cured specimens.

The maximum peaks for composites were located at 94, 92 and 83 °C for the UPR/ZnO400, UPR/ZnO500 and UPR/ZnO600 respectively. The composites show glass transition temperatures higher than the neat UPR, just the UPR/ZnO600 peak was shifted towards lower temperatures than the α signal.

The previous could be associated to the presence of rigid ZnO nanoparticles that restrain the segmental motions of the polyester backbone along the whole crosslinked structure.

### 3.4. Thermal Stability

TGA records the weight loss as a function of temperature, while the derivative thermogravimetric (DTG) curve gives useful information about thermal decomposition stages involved during the overall heating process. Figure 4 shows the TGA and DTG curves obtained from the experimental results performed at a heating rate of 10 °C/min in order to determine the thermal stability of neat UPR and UPR/ZnO nanocomposites. The data extracted from curves are summarized in Table 1 and Table 2.

In this work, the 5% weight-loss was considered as the initial degradation temperature and T_max_ was defined as the temperature at the maximum weight loss rate.

In this work, the thermal oxidative degradation process of UPR was considered to occur in three stages. The initial degradation step was observed at a temperature range of 220–340 °C as resulting from the mass-loss of water dehydration. The second degradation stage occurred in the temperature range of 320–480 °C, and it was ascribed to chain scission of polymer fragments (involving polystyrene and polyester) progressively taking place along the main chain until the fragments are small enough to volatilize, corresponding to a strong peak in DTG curves at around 420 °C. The third stage could be related to the formation of metastable carbonaceous char, which can be further degraded at temperatures around 460 °C. The weight loss of all nanocomposites at the early stages of their thermal degradation might be related to the presence of hydroxyl groups on the ZnO surface that catalyze the thermal degradation of the polyester resin, as observed in the thermal degradation of polyacrylate [38].

In particular, neat UPR and UPR/ZnO400 show similar thermal behavior, suggesting that the addition of 0.05 wt % of ZnO nanoparticles treated at 400 °C did not affect the thermal behavior during the degradation temperature of the polyester. Note the T_onset_ for the UPR and UPR/ZnO400 was 338 and 343 respectively, as listed in Table 1. In addition, both UPR/ZnO500 and UPR/ZnO600 curves show similar T_onset_ values (372 and 369 respectively), which represents about 30 °C higher than the previous one.

The nanocomposites contain similar weight percentages of ZnO nanoparticles with different sizes promoted by the heat treatment. The previous results should indicate the number of well dispersed nanoparticles contained in UPR/ZnO500 and UPR/ZnO600 nanocomposites was higher than neat UPR and UPR/ZnO400, which favored the extent degradation temperature of the polyester.

Thermogravimetric plots obtained from the UPR and the UPR/ZnO nanocomposites at the three heating rates of 5, 10 and 20 °C/min are shown in Figure 5.

Figure 5a shows the TGA curves obtained from the polyester resin. As observed, the initial degradation temperature (T_onset_) values were variable and did not follow any relationship attributed to the heating rate. In contrast, at the final degradation temperature (T_offset_), as the heating rate increased the T_offset_ values also did. In Figure 5b, the TGA curves obtained from the UPR/ZnO400 samples show a similar behavior that was observed in Figure 5a, suggesting that the addition of the ZnO particles treated at 400 °C did not promote a significant change in the degradation temperature of the polyester matrix.

Figure 5c,d depicts the TGA curves corresponding to UPR/ZnO500 and UPR/ZnO600, respectively. The thermograms allowed us to appreciate that the T_onset_ and T_offset_ were related to the increase in the heating rate. This situation might suggest that the decrease in the particle size offers a higher surface area; hence, the heating is homogeneously distributed along the polyester matrix and its decomposition is uniform.

Figure 6 displays the DTG plots corresponding to the UPR and UPR nanocomposites at different heating rates. DTG curves allowed us to reveal the degradation process of the polyester resin and the respective UPR/ZnO nanocomposites seemed to be carried out in a two-step process, instead of a one step process such as previously suggested by the TGA curves. Table 2 presents the transition temperatures obtained from the DTG curves.

### 3.5. Kinetics of Thermal Degradation

The degradation kinetic parameters of the neat UPR and UPR/ZnO composites were calculated using dynamic weight loss data from TGA at the heating rates of 5, 10 and 20 °C/min (Table 1). Figure 7 shows the obtained plots of ln(dα/dt) versus 1/T and ln(1−α) versus 1/T at values of α = 0.2, 0.4, 0.6 and 0.8, and Figure 8 shows the ln(1 − α) versus 1000/T plots The values for the kinetic parameters (E_a_ and n), obtained from the Friedman method, are listed in Table 3. According to the R^2^ from Figure 7a,b, it is possible to detect a low correlation of the experimental data to the Friedman model, especially at the lowest *α* values (0.2 and 0.4). This situation was not observed in Figure 7c,d, where the fitting of the model shows a better correlation for all α values, which was assumed to have uniform degradation of the nanocomposites.

The activation energy, defined as the minimum amount of energy required to initiate the degradation process, can be considered as a quantitative parameter to know the initial thermal degradation behavior of materials [39]. The data in Table 3 show the *E_a_* corresponding to UPR/ZnO400 and UPR/ZnO500 were smaller than neat UPR, which confirmed that the addition of ZnO nanoparticles to UPR produces a catalytic effect that reduces the amount of energy needed to initialize the thermal degradation.

Conversely, Ea corresponding to UPR/ZnO600 was notoriously higher than UPR and the rest of the nanocomposites, which implies the heat treatment applied to the ZnO nanoparticles could promote a stepwise degradation. Therefore, the tendency observed in the E_a_ values seems to be related to the ZnO size reduction. These observations are in agreement with the data obtained from the thermal degradation of ethylene–propylene–diene rubber compounds containing nano-zinc oxide particles [20].

The n value, obtained from the Friedman equation, is directly related to the consumption rate of the reactants in the degradation reaction and could provide an accurate insight into the degradation period of materials.

In this work, the reaction occurred progressively with 0.2 and 0.4 α values, however the reaction changed abruptly at higher values of α, as presented in Table 3. This behavior was associated to the transitions detected during the TGA. At the beginning of the reaction, release of water and CO_2_ took place in major proportions, afterwards the reaction at α values of 0.6 and 0.8 revealed degradation, which resulted from the chain scission of polystyrene and polyester fragments that was accelerated by the catalytic effect of the ZnO particles in the polyester matrix. It is necessary to highlight that the UPR/ZnO400 just revealed one transition peak in the DTG curves meanwhile the rest of materials displayed two signals, as revealed in Figure 6 and reported in Table 2.

Figure 9 shows the curves ln(β/T^2^_p_) versus 1/T_p_ from the Kissinger model, Equation (7). 

Reasonable straight lines were obtained for the entire range of experiments. The Kissinger’s method derives the activation energy using the peak temperature at which the maximum reaction rate occurs (T_p_) and the order of reaction using the shape of the mass loss-time curve. In general, the model represents with high correlation (R^2^ > 0.98) the experimental data, providing evidence of the reliability of this model to depict the degradation characteristics of the studied materials.

Table 4 kinetic parameters obtained from the Kissinger model fitted to the TGA experiments.

Similar to the Friedman model, the kinetic parameters change with the heat treatment applied to ZnO particles incorporated to the polyester resin. The E_a_ and n values, follow similar behavior to the average values obtained from the Friedman equation. Nevertheless, some differences in the numerical values of the kinetic parameters are possible to appreciate. These differences with respect to Kissinger and Friedman are derived from the mathematical basis of both models, as previously described by other authors [15,18,19].

The experimental data were also analyzed using the Coats–Redfern method (Equation (9)), considering different algebraic expressions reported in the literature. The most similar activation energy values (at a heating rate β = 10 °C/min) were found using the Jander equation, $g\left(\alpha \right)={\left[1-{\left(1-\alpha \right)}^{1/3}\right]}^{2}$, indicating the presence of a three-dimensional diffusion mechanism of degradation in the prepared composites [18,19,21,22].

## 4. Conclusions

The ZnO particles were synthetized by means of the sol–gel approach and successfully incorporated to the polyester resin in solution blending at 0.05 wt % concentration, which reduced the molecular mobility of the polyester chains.

TGA experiments were performed at three heating rates of 5, 10 and 20 °C/min in order to determine the degradation kinetic parameters two model-free isoconversional methods: the Friedman model (differential method) and the Kissinger model (integral method) were applied. The activation energy calculated with both methods showed a correlation with the ZnO size, tending to increase when the size of the ZnO decreases, at a lower size (ZnO/600) the high activation energy compared with the blank sample, indicates that the reinforced polymer dissipates more energy and is more resistant to high temperatures.

The evidence found in this work showed that the addition of ZnO particles promoted a catalyst effect that reduced the amount of thermal energy needed to start the thermal degradation.

## Figures and Tables

**Figure 1 polymers-12-01753-f001:**
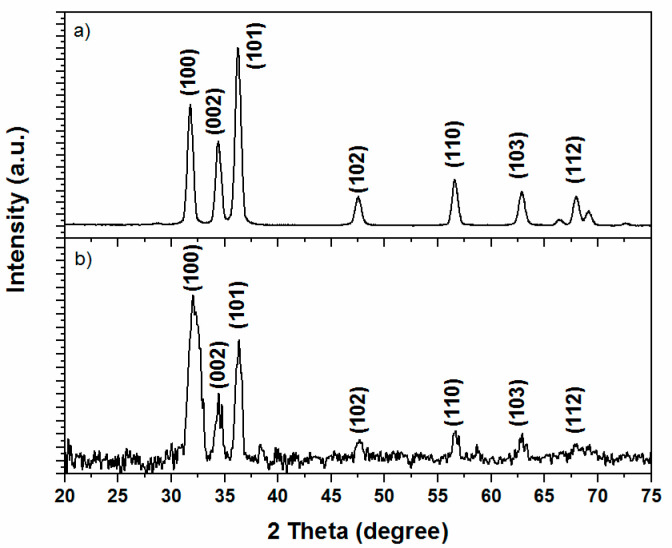
XRD patterns of ZnO particles heat treated at (**a**) 600 °C and (**b**) 400 °C.

**Figure 2 polymers-12-01753-f002:**
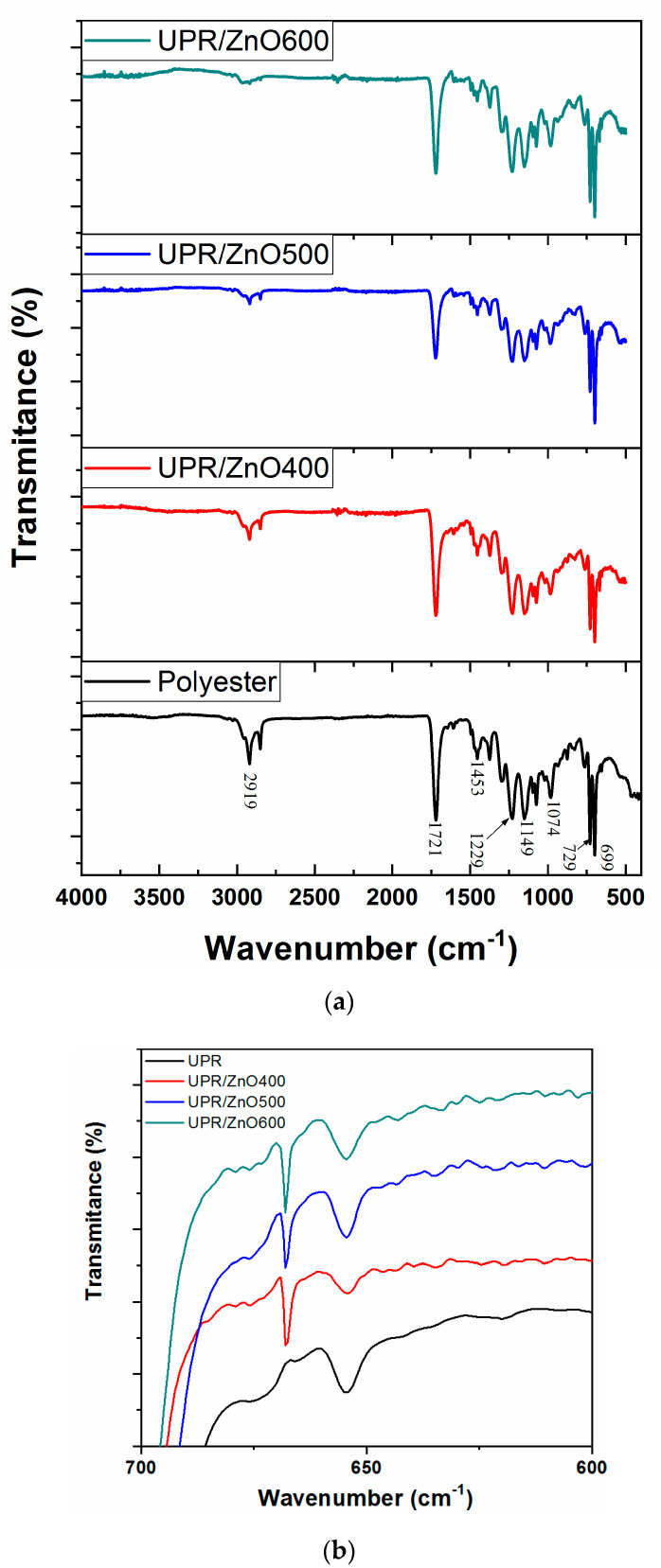
FTIR from (**a**) unsaturated polyester resin (UPR) and nanocomposites and (**b**) a close view from 700 to 600 cm^−1^.

**Figure 3 polymers-12-01753-f003:**
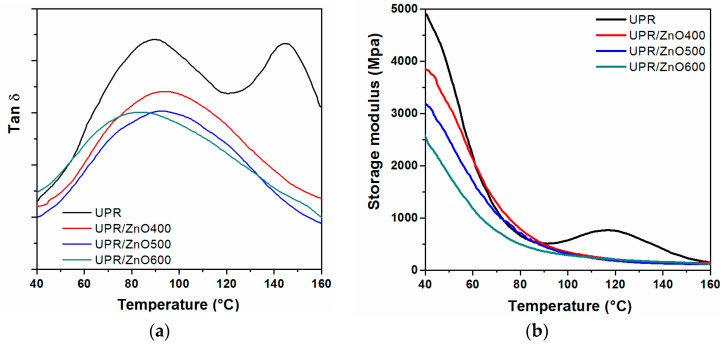
DMA curves from UPR and nanocomposites: (**a**) storage modulus and (**b**) tan δ.

**Figure 4 polymers-12-01753-f004:**
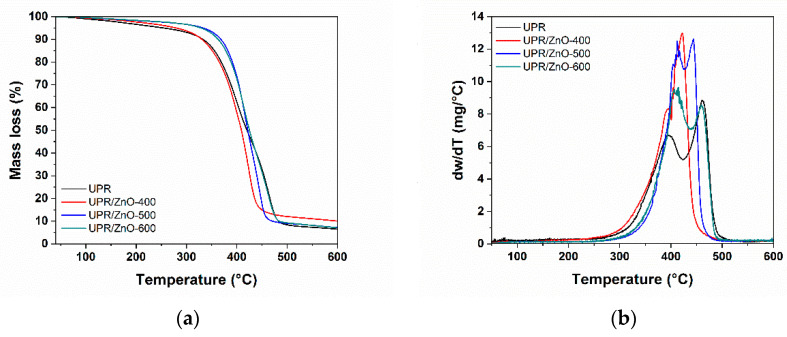
Thermogravimetric plots from UPR and nanocomposites: (**a**) TGA and (**b**) derivative thermogravimetric (DTG).

**Figure 5 polymers-12-01753-f005:**
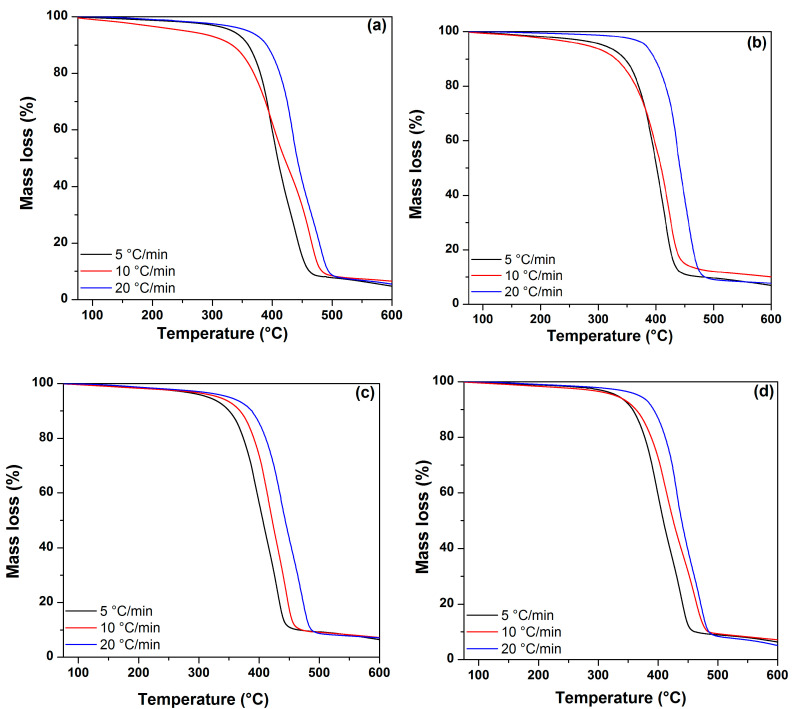
TGA plots from UPR and nanocomposites obtained at heat rate of 5, 10 and 20 °C/min for: (**a**) UPR, (**b**) UPR/ZnO400, (**c**) UPR/ZnO500 and (**d**) UPR/ZnO600.

**Figure 6 polymers-12-01753-f006:**
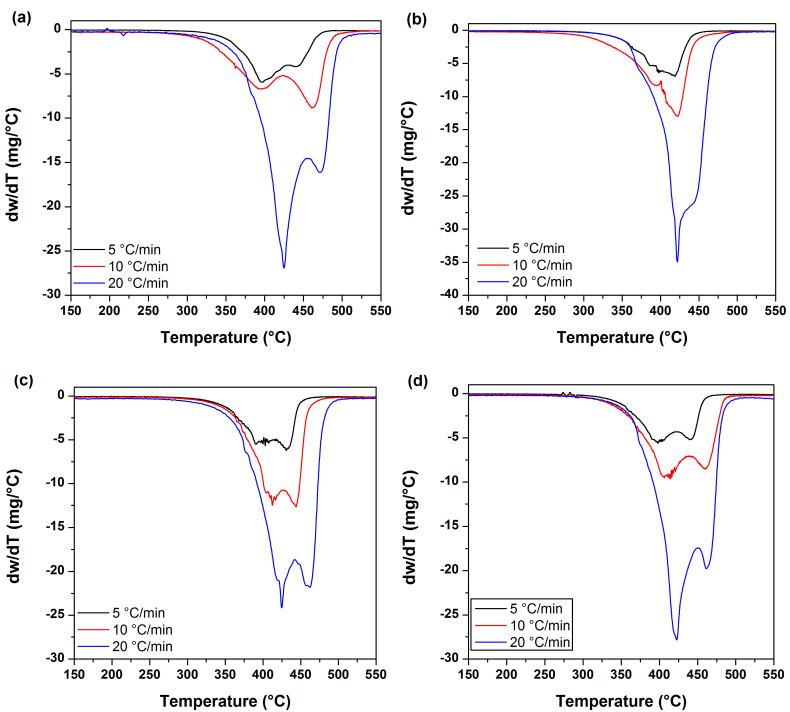
DTG plots from UPR and nanocomposites obtained at a heat rate of 5, 10 and 20 °C/min for: (**a**) UPR, (**b**) UPR/ZnO400, (**c**) UPR/ZnO500 and (**d**) UPR/ZnO600.

**Figure 7 polymers-12-01753-f007:**
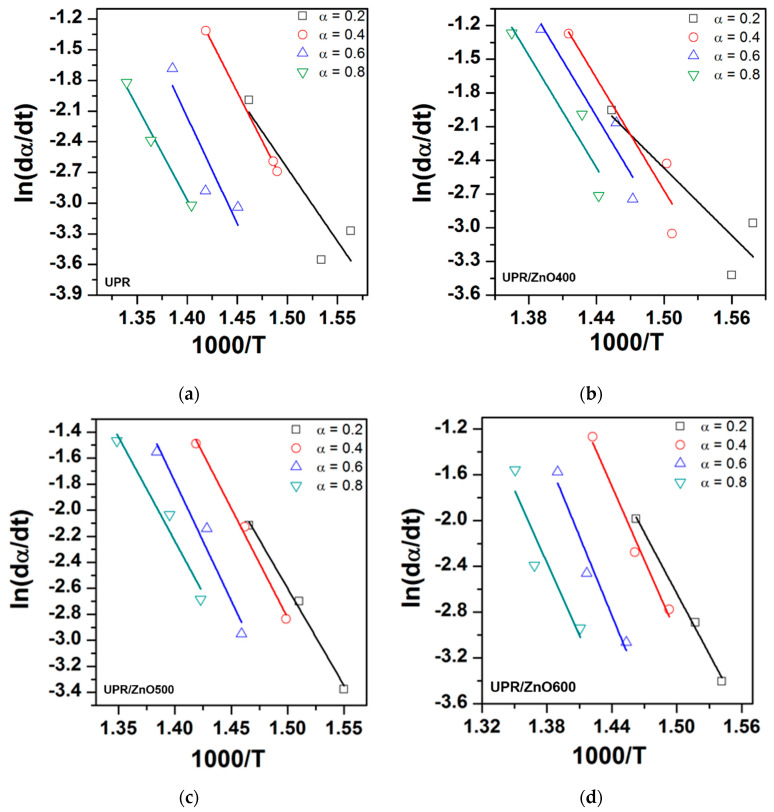
Isoconversion plot of the Friedman method (**a**) UPR, (**b**) UPR/ZnO400, (**c**) UPR/ZnO500 and (**d**) UPR/ZnO600.

**Figure 8 polymers-12-01753-f008:**
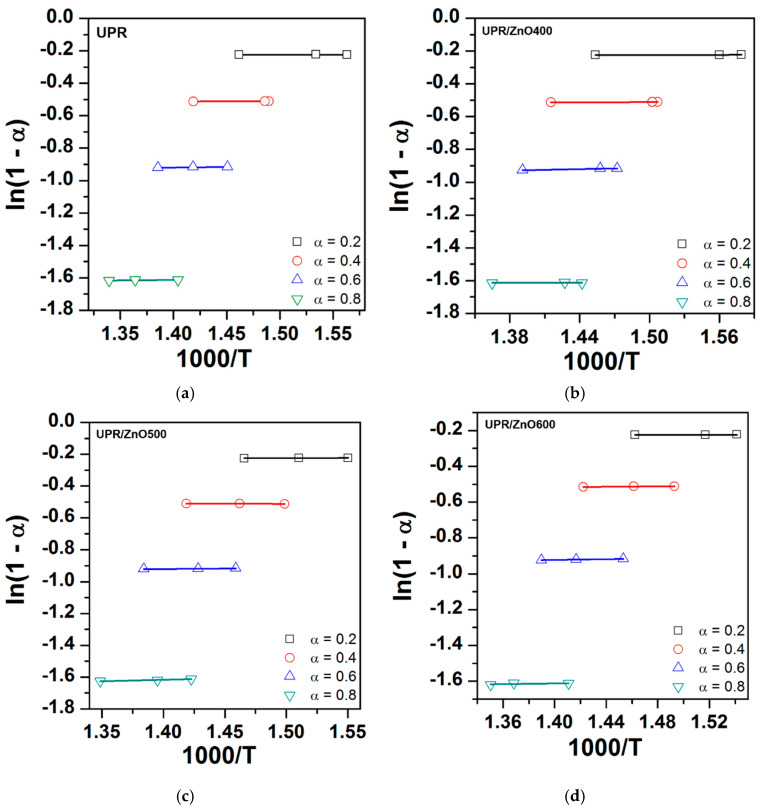
ln(1-α) versus 1000/T plots for (**a**) UPR, (**b**) UPR/ZnO400, (**c**) UPR/ZnO500 and (**d**) UPR/ZnO600.

**Figure 9 polymers-12-01753-f009:**
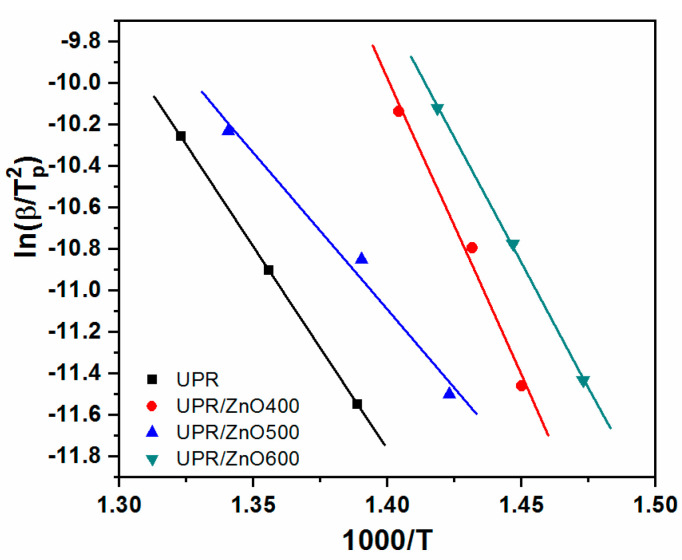
Isoconversion plot of the Kissinger method for UPR and nanocomposites.

**Table 1 polymers-12-01753-t001:** Data obtained from the thermogravimetric curves.

Material	Heating Rate (°C/min)	Temperature (°C)		Degradation (%)		Heat Resistance Index
		Onset	Offset	5	30	
**UPR**	5	363.6	456.6	333.1	391.3	180.3
	10	338.5	463.7	255.1	388.3	164.2
	20	387.1	488.1	359.3	424.6	195.3
**UPR/ZnO400**	5	350.4	434.6	309.5	383.4	173.4
	10	343.1	439.1	280.1	384.2	167.9
	20	381.1	459.6	380.1	428.1	200.4
**UPR/ZnO500**	5	351.9	445.3	315.8	387.1	175.7
	10	371.7	454.8	333.5	404.2	184.2
	20	389.9	474.7	350.6	424.3	193.5
**UPR/ZnO600**	5	357.2	453.3	332.1	389.7	179.6
	10	369.5	476.9	329	403.2	183.1
	20	388.8	476.9	368.3	423.3	196.6

**Table 2 polymers-12-01753-t002:** Transition temperatures obtained from the DTG curves.

Material	Heat Hate (°C/min)	Temperatures (°C)	
		Transition 1	Transition 2
**UPR**	5	395.1	440.1
	10	399.2	465.1
	20	424.6	470.6
**UPR/ZnO400**	5	416.6	-
	10	425.4	-
	20	432.5	-
**UPR/ZnO500**	5	390.9	429.6
	10	416.1	445.9
	20	434.2	467.6
**UPR/ZnO600**	5	399.7	439.6
	10	411.4	461.2
	20	430.7	469.9

**Table 3 polymers-12-01753-t003:** Kinetic parameters E_a_ and n for different values of α, at the right of each parameter is the corresponding correlation coefficient R2 obtained from the data fit with the Friedman model.

Material	*α*	E_a_	R^2^	n	R^2^
**UPR**	0.2	118.66	0.81	0.34	0.06
	0.4	159.62	0.99	0.66	0.96
	0.6	173.38	0.84	3.43	0.72
	0.8	151.33	0.98	3.08	0.71
	Average	150.75		1.88	
**UPR/ZnO400**	0.2	83.05	0.81	0.49	0.99
	0.4	138.51	0.91	0.76	0.99
	0.6	138.91	0.91	4.28	0.96
	0.8	138.32	0.88	3.37	0.05
	Average	124.69		2.23	
**UPR/ZnO500**	0.2	124.19	0.99	0.32	0.77
	0.4	139.74	0.99	0.76	0.80
	0.6	151.88	0.96	3.51	0.99
	0.8	132.81	0.96	3.71	0.97
	Average	137.15		2.13	
**UPR/ZnO600**	0.2	147.14	0.99	0.76	0.79
	0.4	178.66	0.99	2.84	0.97
	0.6	191.86	0.96	4.36	0.92
	0.8	175.19	0.97	4.25	0.31
	Average	173.19		3.05	

**Table 4 polymers-12-01753-t004:** Degradation kinetic parameters obtained from the Kissinger equation.

Material	E_a_	n	R^2^
**UPR**	128	0.66	1.0
**UPR/ZnO400**	139	0.63	0.987
**UPR/ZnO500**	151	3.52	0.987
**UPR/ZnO600**	191	2.85	0.99

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
