# Peer review of "Thermal Degradation Kinetics of ZnO/polyester Nanocomposites"

_polymers, 2020, doi:10.3390/polym12081753_

Round 1

Reviewer 1 Report

Comments:

In this manuscript entitled “Synthesis and characterization of ZnO/polyester nanocomposites and their thermal degradation kinetics”, the authors claimed that ZnO synthesis via sol-gel method and fabrication ZnO/polyester nanocomposite. Authors also claimed based the Friedman and Kissinger models, the kinetic mechanism (such activation energy and reaction order) can be predicted by thermosgravimetric analysis data.  However, authors only used few sentences to describe the synthesis of nanoparticle and fabrication of nanocomposite and the conclusions of kinetic analysis result not consistent with each other. It makes readers is hard to understand this manuscript. Authors need to make some changes in the conclusions of kinetic analysis and adding more detail information of synthesis nanoparticle and fabrication nanocomposite for reader to easier understand this manuscript.      

1. The title of this manuscript is “Synthesis and characterization of ZnO/polyester nanocomposites and their thermal degradation kinetics”. However authors only briefly described the synthesis nanoparticle and fabrication nanocomposite which makes this manuscript main focus on the kinetic mechanism analysis. Authors need to more detail to describe how to control the properties of nanoparticle and nanocomposite for optimizing the performance of nanocomposite. Or authors need to use the other title.

2. In the introduction section, authors inappropriate cite references. For example in line 47-51 “first two processes deal with surface treatment, where bulk properties like mechanical or thermal cannot be achieved”. In the reference 12, the mechanical or thermal properties is for nanocomposite not for ZnO nanoparticles. Authors need to re-check the references for appropriate citation in the manuscripts.        

3. For theory of kinetic analysis (line 138-177), this is hard to understand how to receive activation energy (Ea) and reaction order (n) from the simple plot without any assumption. For example, authors used equation (5) and draw the ln ( ) vs 1/T plot and claimed the slope is Ea/R, however the n and (the fraction of solid degraded) in ) term both depend on the temperature (T). why authors can use the linear relationship to fit the data points and use the slope to calculate the activation energy? Authors need to add more detail information for kinetic analysis to solidify their claims.

4. Some discussions are ambiguous, authors need to clarify them

  • In line 200-201, authors used the Scherrer’s equation to calculate the average particle size based on XRD results. Authors need to add more detail information about how the nanoparticle size is analyzed based on the special peak’s FWHM or average of all peaks’ FWHM.
  • In Figure 1, in order to prove the crystal structure of as-made ZnO particle, authors need to add the information of reference peaks or cite the standard XRD files (such as JCPDS file 36-1451 of ZnO).
  • In line 205-206, authors claimed that ”The prominent bands of Zn-O bond typically observed at around 428 cm−1 resulting to the stretching vibration bond in tetrahedral coordination”. But the FTIR spectrum in figure 2 only with the range from 4000 cm-1 to 500 cm-1. Please add the spectrum with wavenumber < 500 cm-1 to confirm this claim.
  • In 212, unsaturated (-C=H) stretching vibration is not correct. Please re-check the chemical structure.
  • Line 216, “It seems the heat treatment on ZnO fillers is promoting the presence of –CH=CH- groups, which are responsible the bands almost disappears in the IR spectrum of  ” The IR spectrum of CH=CH- characterization increase with incorporation of ZnO which result from the presence of CH=CH- on ZnO fillers. Please explain why “almost disappears” of IR spectrum.
  • In line 232, “Close observations in Figure 2b allows to appreciate relevant bands present in 232 nanocomposites at 668 cm−1. The signals are assigned to the stretching vibrations of Zn-O bonds in octahedral arrangements and further confirms the wurtzite structure” please add the reference to prove this claim.
  • In line 248, “ the heat treatment produced an effective size reduction of the ZnO particles, reducing the surface area as well and leads to the low stiffening effect”. This claim is not correct. The surface area/volume ratio increase with decrease the size of nanoparticle. Based on doping density of ZnO nanoparticle in polyester is 0.05 wt% of ZnO, the contact surface of nanoparticle and polyester increase with decreasing particle size 
  • In line 280 and 285, “ The composites show glass transition temperatures higher than the neat UPR” and “Similar results had been observed for other authors when adding small concentrations of nanoparticles, concluding that the reduction of stability thermal-mechanical is due to weak chemical interaction between nanoparticles and polyester matrix.” These two sentences are not consistent to each other. Please re-check these claims. And please add the reference the second sentence.
  • Line 332 and line 335, “disordered decomposition of the polyester resin and the UPR/ZnO400 composite” and “uniform degradation of the nanocomposites”. Please direct information (such as SEM image) to confirm these claims.

Author Response

Authors want to thank you for your valuable comments and recommendations

Authors covered point by point your suggestions that rich the manuscript

Reviewer 2 Report

Authors report an interesting material but the presentation is not lucid. The following issues need to be addressed before the manuscript may be accepted for publication:

  1. The authors report the thermal degradation kinetics of as-prepared composite material. However, the manuscript is unable to show the relevance of the study. I suggest authors to clearly state the relevance and novelty of their work in ‘introduction’ or other appropriate section.
  2. The manuscript lacks some basic morphological characterization of the prepared samples. I suggest the authors to include SEM micrographs with their justifiable elaboration so that the formation of the desired product can be substantiated.
  3. The form in which the authors have cited references in the text doesn’t match with the standard format of ‘polymers’ journal. It needs to be corrected. For instance, in line 32, the references should be enclosed within square brackets, i.e., it should be [1,2]. The same format should be applied in the whole manuscript wherever applicable.
  4. (Materials and methods, line 107-115). Please reorganize/supplement the information to clarify the intended meaning of the procedure. The following information is not clear: Why was a single concentration (0.05 wt%) of the ZnO used to obtain the composites? What could be the changes in the properties of composite if other concentrations were used? Did ZnO form a ‘solution’ while mixing with ethanol or it was a ‘suspension’? (Authors have correctly phrased “ZnO nanoparticles were suspended …” in line 110 but used the term ‘solution’ vaguely in other places. Authors should keep in mind the difference between suspension and solution).
  5. (Section 2.5 Kinetic models). The theoretical information provided in this section seems to be more conclusive. I suggest authors to include more elaboration pertaining to the relevance of the study. For instance, please add a sentence on what parameters are important for studying degradation behavior (line 138), add a sentence on why there is need of Kissinger method (line 162), etc. Please maintain a systematic link between the preceding and following sentences and/or paragraphs. An article (among many others) has been suggested for the reference so that the authors may improve the manuscript: Coatings 2020, 10, 413; doi:10.3390/coatings10040413.
  6. (Lines 191/192, Figure 1, XRD). The manuscript contains the XRD spectra of ZnO particles thermally treated at 400 and 600°C only. I suggest authors to include the spectra of ZnO thermally treated at 500°C also. Furthermore, the X-axis name is oddly abbreviated and the grid lines in Y-axis are unusually numerous. What could be the reason behind this?
  7. (Lines 180-196). The explanation provided for XRD spectra contains a good sort of information but is not well organized. A reference has been advised for organized elaboration of the spectra: Catalysts2019, 9(6), 498; doi:10.3390/catal9060498. Please also mention the ZnO peaks that are seen on either side of (112) plane. Instead of using the terms ‘upper’ and ‘lower’ in caption, it is better to assign sub-number such as a, b.
  8. (Line 236, Figure 2). There is no consistent use of grid lines in the figures. Specifically, there are right Y and top X grid lines in ‘figure a’ but not in ‘figure b’. The grid lines are facing outward in ‘figure b’ but inward in ‘figure a’. In Figure 4, the axis lines used in the figures are too thin in comparison to those in other figures, and so on. I suggest authors to maintain consistency in the format of figures in whole manuscript. In caption of Figure 2: should there be IR only or FT-IR?
  9. (Line 238, parenthesis). The term ‘tan 8’ seems to be incorrectly used. Please rethink the sense of using it in the textual context. In line 134, 6000 °C seems to be wrongly typed because the corresponding TGA curve shows the temperature up to 600 °C only.
  10. The TGA curve presented in Figure 4 (a) has indexed UPR twice but does not show spectra of UPR/ZnO-400. What can be the reason behind this? Additionally, I suggest authors to add degradation conversion profiles of each sample separately at different heating rates (5, 10 and 20 °C/min).
  11. (Line 298/299, Table 1). The table needs to be revised to include more relevant information. Authors can take reference of the same article as advised above. Table 2: please explain in the text why Pol/ZnO400 does not have transition 2 values. Table 3: please clarify how two different R2 values are involved.
  12. (Line 416-506) References are not organized in the correct format of ‘polymers’ journal. I suggest the authors to arrange the references based on the format of this journal.
  13. There are many claims/explanations in the manuscript that need experimental evidence or proper citations. For instance, experimental evidence or proper citations are essential for the claims in lines 57-59, 68/69, 70-73, 138, 223-225, 227/228, 233/234, 265-267, 277/278, 285-287, 291-293, 300-302, 303-306, 307/308, 320-323, 331-333, 334/335, 351-353, and 379/380.
  14. The sample materials are abbreviated in three different ways in the Figures, text and Tables. Are the materials illustrated in Figures different from those written in Tables? Is UPR/ZnO-400 same or different from UPR/ZnO400 and Pol/ZnO400?
  15. (Lines 384-410, Conclusion section). There are unusual and unnecessary changes of paragraphs; some paragraphs consist of a single sentence only. Please reorganize ‘Conclusion’ in a single paragraph maintaining a smooth and systematic link between the sentences.
  16. There are numerous language related mistakes and typos in the manuscript. Some words (e.g. appreciate) are redundantly used while the others are inappropriate in the mentioned context. Many paragraphs are unusually short; they contain one or two sentences only. Some sentences are unable to deliver the intended meaning. The smooth link between the paragraphs is missing in some places. Hence, correction of typos (e.g. subscripts, superscripts, periods, misspelling, etc) and English language editing of the manuscript is recommended.

Author Response

(The authors gave the same response as above.)

Round 2

Reviewer 1 Report

After the revised, this manuscript is more easier for reader to understand.   

Reviewer 2 Report

I am satisfied with the revision and recommend its publication.